# Enhancing Semi-Supervised Semantic Segmentation of Remote Sensing Images via Feature Perturbation-Based Consistency Regularization Methods

**DOI:** 10.3390/s24030730

**Published:** 2024-01-23

**Authors:** Yi Xin, Zide Fan, Xiyu Qi, Ying Geng, Xinming Li

**Affiliations:** 1Key Laboratory of Target Cognition and Application Technology, The Aerospace Information Research Institute, Chinese Academy of Sciences, Beijing 100190, China; xinyi20@mails.ucas.ac.cn (Y.X.); qixiyu20@mails.ucas.ac.cn (X.Q.); gengying@aircas.ac.cn (Y.G.); 2The Aerospace Information Research Institute, Chinese Academy of Sciences, Beijing 100190, China; 13911729321@139.com

**Keywords:** remote sensing, semantic segmentation, semi-supervised learning, consistency regularization, feature perturbation, contrastive learning

## Abstract

In the field of remote sensing technology, the semantic segmentation of remote sensing images carries substantial importance. The creation of high-quality models for this task calls for an extensive collection of image data. However, the manual annotation of these images can be both time-consuming and labor-intensive. This has catalyzed the advent of semi-supervised semantic segmentation methodologies. Yet, the complexities inherent within the foreground categories of these remote sensing images present challenges in preserving prediction consistency. Moreover, remote sensing images possess more complex features, and different categories are confused within the feature space, making optimization based on the feature space challenging. To enhance model consistency and to optimize feature-based class categorization, this paper introduces a novel semi-supervised semantic segmentation framework based on Mean Teacher (MT). Unlike the conventional Mean Teacher that only introduces perturbations at the image level, we incorporate perturbations at the feature level. Simultaneously, to maintain consistency after feature perturbation, we employ contrastive learning for feature-level learning. In response to the complex feature space of remote sensing images, we utilize entropy threshold to assist contrastive learning, selecting feature key-values more precisely, thereby enhancing the accuracy of segmentation. Extensive experimental results on the ISPRS Potsdam dataset and the challenging iSAID dataset substantiate the superior performance of our proposed methodology.

## 1. Introduction

In recent years, remote sensing image processing has become an important research area, among which the semantic segmentation of remote sensing images is a key task [1,2,3,4]. The goal of semantic segmentation is to assign each pixel in the image to an interpretable category. At present, the semantic segmentation of remote sensing images has played a significant role in fields such as military reconnaissance [5], urban planning [6], and environmental monitoring [7], greatly facilitating the automation and decision optimization in these areas. However, due to the characteristics of remote sensing images, conducting pixel-level manual annotation is extremely laborious and time-consuming. Therefore, semi-supervised semantic segmentation has become a major research direction to reduce the cost of manual annotation.

The goal of semi-supervised semantic segmentation is to achieve high-precision segmentation results with a smaller proportion of labelled images. The main types of semi-supervised semantic segmentation methods currently include methods based on consistency regularization [8,9,10,11,12], methods based on pseudo-labels [13], and the recently popular methods based on contrastive learning [14,15,16]. The core of methods based on consistency regularization is to add perturbations to the input, and to and train the model with the aim of maintaining output consistency. Research on consistency regularization models in natural scenes is relatively comprehensive. However, remote sensing scenarios differ significantly from natural scenes. Particularly, the foreground types in remote sensing images are more complex and densely distributed, and the differences between the categories are minimal, leading to a relatively chaotic feature distribution. Currently, the consistency regularization methods applied in the field of remote sensing are mostly extensions of methods from natural scenes. For instance, Zhang [7] applies various random transformations and perturbations to images and predicted labels; ICNet [17] switches between student and teacher networks based on training rounds, adding network perturbations. Although these methods have improved the results of the remote sensing dataset, they focus more on the image level and do not consider the feature level of the remote sensing images. We believe that optimizing the feature level of remote sensing images is equally important.

For remote sensing images, by analyzing their features, we find that the feature distribution is more scattered and chaotic compared to natural scenes. However, existing consistency regularization methods for remote sensing images do not pay much attention to the feature level. To address this, we aim to add feature-level perturbations to force the model to optimize the feature space. In the Mean Teacher (MT) [8] model, the teacher network assists the student network, requiring consistency in the prediction maps generated by different image enhancements. Considering the feature level of remote sensing images, adding different feature perturbations in the student and teacher networks to implement a consistency regularization strategy is an effective solution for feature optimization. At the feature level, after perturbing image features, a loss function must be chosen to evaluate the consistency of the prediction results of the two networks, thus aiding in with updates. The commonly used cross-entropy loss clearly does not meet our needs. Contrastive learning, essentially targeting the feature space, is obviously a good method. There are already corresponding remote sensing semantic segmentation applications using this method [6], achieving decent results. However, traditional contrastive learning still faces the problem of unreasonable negative sample selection. Therefore, we propose a new strategy, setting an entropy threshold to filter key values, to address the problem of more dispersed feature distribution caused by the unique characteristics of remote sensing images.

In this paper, considering the existing problems of current methods, we propose a novel semi-supervised semantic segmentation framework that combines consistency regularization and contrastive learning. For consistency regularization, based on MT, in addition to image-level disturbances, we introduce feature-level disturbances, allowing the model to pay more attention to the feature level of the image during the learning process, compared to previous methods. To achieve this goal, we choose to use contrastive learning to calculate the contrastive loss, assisting in the optimization at the feature level, and completing the final network parameter updates. For the key selection used in contrastive learning, we use entropy thresholds to select positive keys and negative keys, making the results of contrastive learning more precise.

The main contributions of this paper can be summarized as follows:We have introduced a new semi-supervised segmentation framework for remote sensing images. The framework integrates consistency regularization and contrastive learning, enhancing the disturbances at the data and feature levels, and improves feature classification performance through contrastive learning. In addition, this method achieves state-of-the-art performance in popular segmentation benchmarks.We proposed a new consistency regularization method based on MT [8]. By enhancing perturbations at the feature level, the difficulty of maintaining the consistency of image features increases, thus adding to the training difficulty and improving the generalization ability of complex images. Feature perturbations plays a key role in this process and help the model to learn from more challenging features.We utilize contrastive learning at the feature level to achieve a better divide and category selection for the features. A threshold for entropy is established to aid in feature selection, sifting out more accurate negative samples.

## 2. Related Works

### 2.1. Supervised Semantic Segmentation

Semantic segmentation serves as a cornerstone operation in the realm of computer vision. FCN [18] is groundbreaking in the field of semantic segmentation. By replacing fully connected layers with convolutional layers, FCNs extend convolutional neural networks from image classification tasks to pixel-level prediction tasks. FCNs also introduce an upsampling operation to obtain fine segmentation results. Despite some limitations, FCN lays the foundation for subsequent research, as mentioned earlier. U-Net, primarily used for medical image segmentation, adopts a symmetric encoder-decoder structure and achieves feature fusion through skip connections. This design allows U-Net [19] to fully utilize multi-scale features, hence yielding excellent performance in segmentation tasks. DeepLabV3+ [20] is the latest in the DeepLab series that is focused on studying dilated convolution and Atrous Spatial Pyramid Pooling (ASPP). Building on the foundation of DeepLabV3, DeepLabV3+ introduces a decoder module to achieve finer segmentation results. PSPNet [21], through the introduction of a pyramid pooling module, effectively captures the context information of different scales. This capability enables PSPNet to achieve accurate semantic segmentation in multi-scale scenes. Fast-FCN [22] is an efficient semantic segmentation method. By introducing a global average pooling layer, Fast-FCN achieves the rapid aggregation of features. Compared with traditional FCN, Fast-FCN significantly improves running speed while maintaining accuracy. For different scenarios, some models have improved their specific structures to adapt to specific tasks. For example, PSNet [23], addressing the issue of low detection rate and a high false alarm rate in forest fire smoke detection, introduces a detail-difference-aware module to distinguish between smoke and smoke-like objects, and an attention-based feature separation modules to suppress interference features.

In addition, methods based on the Transformer architecture have also achieved very good results. For instance, Segmenter [24], which is based on the Transformer architecture, allows the model to capture global context information in both the first layer and throughout the entire network. TransUNet [25] also employs a hybrid visual Transformer as an encoder for stronger feature extraction, and it has achieved state-of-the-art results in medical image segmentation. SegFormer [26] is also a simple, efficient, and powerful semantic segmentation framework that combines Transformers with a lightweight multi-layer perceptron decoder.

### 2.2. Semi-Supervised Semantic Segmentation

Common strategies in semi-supervised semantic segmentation involve leveraging the principle of consistency regularization [9,27,28,29], which is also used in semi-supervised classification tasks. In this regard, CutMix [10], MixUp [30] extend unlabelled data and enforce consistency under these deformations. Another strand of consistency regularization philosophy involves transferring knowledge from labelled to unlabelled data. The Mean Teacher [8] is utilized to enforce consistency for the predictions of unlabelled images across different training epochs. Similarly, the work in [31] introduced a technique that maintains the exponential moving average of previous models using self-ensembling to achieve stable outputs for unlabelled data. To increase network perturbations, PSMT [32] employs a dual-teacher network to assist in the training and optimization of the student network.

Pseudo-label-based methods constitute another mature research direction in semi-supervised semantic segmentation. These methods rely more heavily on prior predictions made on unlabelled data and extend the dataset using these predictions. A typical representative example is self-training methods, where ST++ [33] applies data augmentation techniques to unlabelled images during the self-training process.

Another method based on adversarial learning [34,35] has also shown promising results, such as in [36], where the discriminator’s task is to distinguish unlabelled image segmentation outputs originating from the labelled or unlabelled pools, forcing the generator to create indistinguishable segmentation between the two.

Contrastive learning, which emphasizes high-level features, enables the network to distinguish classes well without real labels. There have been some works on semantic segmentation using contrastive learning [37,38,39,40]. Reco [15] was the first to apply pixel-level contrastive learning to the field of semantic segmentation. Moreover, Chen et al. [41] only samples positive examples when using contrastive learning. Lai et al. [42] adopts a self-supervised learning paradigm, selecting reliable pixel points from differently augmented images as positive and negative samples for contrastive learning.

### 2.3. Semi-Supervised Semantic Segmentation of Aerial Imagery

Remote sensing images have high resolution, with the number of pixels in a single image far exceeding those in other fields. At the same time, the categories in remote sensing images are more complex, making manual pixel-level annotation work more time-consuming and laborious. To address this labor-intensive annotation issue, in recent years, some researchers have started to conduct research on semi-supervised semantic segmentation. Zhang et al. [7] performed Transformation Consistency Regularization on the prediction labels of the teacher network and compared the results with the student network, extending randomness to the label level. ICNet [17], during the training process, alternates between transforming the student network and the teacher network. This approach allows the two networks to supervise each other, thereby increasing perturbations at the network level. PICS [5] adopts a selective self-training strategy. By using labelled images, it selects generated samples that are closer to the true values, thereby reducing the accumulation of potential errors. UniMatch [43] improved the structure of FixMatch and performed Dropout operations on the features after the decoder, introducing additional disturbances at the feature level to achieve better segmentation results. Kerdegari et al. [44] used a GAN network to enlarge the annotated dataset. Yang et al. [6] improved the final segmentation accuracy using contrastive learning. Zhang et al. [45] combined the CPS structure and integrated prediction results to adapt to SAR image semantic segmentation. Fang [46] introduced a clustering algorithm into the co-learning algorithm, generating high-quality pseudo-labels by integrating features. Hong [47] utilized multiple types of data, SAR images, and multispectral images, extracted more features to assist in pixel classification; and redesigned a multimodal classification framework for this purpose.

Compared to the direct transfer of methods from natural scenes, these works improve the segmentation effect of remote sensing images significantly. However, the attention received by the semi-supervised semantic segmentation of high-resolution remote sensing images is far from matching its practical importance. Inspired by the consistency regularization method and the contrastive learning method, and considering the characteristics of remote sensing images, we have rethought how to truly specialize semi-supervised semantic segmentation for remote sensing images.

## 3. Materials and Methods

### 3.1. Methods

In this section, we will elaborate on our research methodology in detail. Our approach is designed and optimized based on the Mean Teacher architecture. Mean Teacher, as an effective semi-supervised learning method, primarily aims to reduce the model’s sensitivity to minor variations in input data through consistency regularization, thereby enhancing the model’s generalization capability. Building on this, we have made a series of improvements to the network structure to cater to our specific tasks and data. To perturb the features, we have added different feature perturbation modules after the encoder in both the student and teacher networks. Simultaneously, for the progression of contrastive learning, we have added a feature representation head parallel to the decoder to adapt to our specific task and data. The specific network structure and model flow are shown in Figure 1. First, we set up a semi-supervised dataset.

Labelled dataset: $D_{l}={\left\{\left(x_{i}^{l},y_{i}^{l}\right)\right\}}_{i=1}^{|D_{l}|}$

Unlabelled dataset: $D_{u}={\left\{\left(x_{i}^{l}\right)\right\}}_{i=1}^{|D_{u}|}$

According to the principle of semi-supervised learning, labelled images represent only a small fraction of all training images, $|D_{l}\left|\ll \right|D_{u}|$. The proposed framework consists of a student network and a teacher networks. The student and teacher networks have the same architecture but different parameters, where the parameters $\theta_{t}$ of the teacher network is the exponential moving average (EMA) of the student network parameters $\theta_{s}$. The update formula for $\theta_{t}$ is as follows:(1)$$\theta_{t}^{e}=\gamma \theta_{t}^{e-1}+\left(1-\gamma \right)\theta_{s}^{e}$$

γ is the smoothing factor, γ∈(0,1). *e* represents the training epoch.

Our method optimizes the prediction classification at both the image level and feature level by targeting the features of remote sensing images. Through learning at different levels, effective image segmentation is achieved. The overall framework is divided into two parts, each with the following structure:

Feature Disturbed Mean Teacher Model (FDMT): FDMT is a new paradigm for semi-supervised semantic segmentation. Building upon the traditional Mean Teacher module, it additionally incorporates a feature disturbance component. This enables the multi-level optimization of pixel classification in images, leading to improved classification results.

Contrastive Learning with Entropy Threshold Assisted Feature Sampling: This paper utilizes contrastive learning to aid with the optimization of the feature space. In conducting contrastive learning, we employ entropy as an auxiliary means for sampling queries, positive keys, and negative keys. By setting an entropy threshold, we aim to filter out more accurate key values, thereby facilitating more efficient contrastive learning.

#### 3.1.1. Feature Disturbed Mean Teacher Model (FDMT)

In the method proposed in this study, the model comprises a teacher network and a student network. Unlike the network structure of traditional Mean Teacher, each network includes an encoder head *h*, a decoder head *f*, and a representation head *r*. After the encoder, a feature perturbation module is added to introduce feature perturbations. The reason for adopting an additional representation head here is as follows: The features for contrastive learning need to be more general, and the features directly obtained from the encoder and decoder may not yield the best results. Therefore, an additional representation head is designed to extract and contrast features, enhancing their discriminability. For this, we use the representation head *r* designed in [15]. After each round of training, the parameters of the student network are updated based on the loss.

During each training iteration, we carry out a random sampling to obtain an identical number of labelled images $N_{l}$ and unlabelled images $N_{u}$, with $|N_{l}\left|=\right|N_{u}|$. For each labelled image, we input it into the student network to make predictions, and then we compare the predicted labels with the ground truth labels to calculate the supervised loss $L_{sup}$.
(2)$$L_{sup}=\frac{1}{|N_{l}|}\sum_{\left(x_{i}^{l},y_{i}^{l}\right)\in N_{l}}l_{ce}\left(y_{si}^{l},y_{i}^{l}\right)$$
(3)$$y_{si}^{l}=O\left(S\left(f\circ h(x_{i}^{l};\theta_{s})\right)\right)$$

$l_{ce}\left(•\right)$ represents the cross-entropy loss, and $y_{i}^{l}$ denotes the manually annotated labels. S(•) represents the softmax function, and O(•) represents the one-hot encoding form. S(f∘h) represents the segmentation probability map generated after the image passes through the encoder *h* and decoder *f* successively. $y_{si}^{l}$ is the final one-hot encoded form. $\theta_{s}$ denotes the parameters of the student network.

For unlabelled images, our process differs significantly from the Mean Teacher model. First, let’s discuss the student network part. The same unlabelled image follows two different processing streams: one with strong data augmentation, where we use Cutmix, and one without data augmentation. Here, the use of strong image augmentation is designed for perturbations at the image level, while the process without image augmentation aims for a more rational and controllable addition of feature perturbations. By doing so, we simultaneously introduce disturbances at both the image and feature levels, thereby optimizing the model on multiple dimensions. The images, having gone through different augmentation processes, enter the student network, generating two prediction labels, namely $y_{s1}^{u}$ and $y_{s2}^{u}$. For images that have not undergone image augmentation, after entering the encoder, a VAT perturbation δ is added to them, which is defined as follows:(4)$$F_{s2}^{u}=h\left(x^{u}\right)+\delta$$
(5)$$\delta =\arg\;\max_{||\delta ||\leq \epsilon }\;D_{KL}[S\left(f\left(h\left(x^{u}\right);\theta_{s}\right)\right)||S\left(f\left(h\left(x^{u}\right)+\delta ;\theta_{s}\right)\right)]$$
(6)$$y_{s2}^{u}=O(S\left(f\left(F_{s2}^{u}\right)\right)$$

In the above formula, δ represents the Virtual Adversarial Training (VAT) [48] perturbation generated by the student network. $S(f\left(h\left(x^{u}\right);\theta_{s}\right))$ represents the the softmax probability of predicted label generated by the image without image perturbation and feature perturbation through the student network. $S(f\left(h\left(x^{u}\right)+\delta ;\theta_{s}\right))$ is the softmax probability of a predicted label generated after the image without image augmentation is augmented with VAT perturbation. $D_{KL}\left[•\right]$ represents the Kullback-Leibler divergence, which is a measure of the difference between two probability distributions.

As for the teacher network, the same unlabelled image is also divided into two processing steps: weak data augmentation and no augmentation. After the two images pass through the encoder *h*, it generates two sets of image features, $F_{t1}^{u}=h\left(x^{u};\theta_{t}\right)$ and $F_{t2}^{u}$. We pass $F_{t1}^{u}$ directly through the decoder *f*, i.e.,: $y_{t1}^{u}=O(S\left(f\left(F_{t1}^{u}\right)\right)$. For $F_{t2}^{u}$, we simply jitter its features; specifically, we add extremely weak uniform distribution noise to the features. Here, we do not use the same processing method as the student network, because the VAT perturbation is undoubtedly a strong perturbation at the feature level. Here, we have transferred the idea of image-level perturbation, where the prediction of weak perturbation assists in optimizing the prediction of strong perturbation. Therefore, we chose a feature jittering method in the teacher network that has less impact on the features. After adding the feature perturbations and passing it through the decoder, the prediction map $y_{t2}^{u}=O\left(S\left(F_{t2}^{u}\right)\right)$ is generated.

After the teacher and student networks generate their respective prediction maps, different types of losses are computed. For the unsupervised loss part, we use the prediction maps $y_{s1}^{u}$ and $y_{t1}^{u}$, with the unsupervised loss being defined as follows:(7)$$L_{unsup}=\frac{1}{|N_{u}|}\sum_{\left(x_{i}^{u}\right)\in N_{u}}l_{ce}\left(y_{s1i}^{u},y_{t1i}^{u}\right)$$

For $y_{s2}^{u}$ and $y_{t2}^{u}$, we will perform contrastive learning calculations. The specific calculation steps will be given in the next section.

#### 3.1.2. Contrastive Learning with Entropy Threshold Assisted Feature Sampling

Contrastive learning, initially utilized in image classification tasks, aims to define the query, the positive key, and the negative key. It seeks to learn the similarity between the query and the positive key, while simultaneously reinforcing the difference between the query and the negative key. As this method is incorporated into semantic segmentation, the sample scope transitions from image-level to pixel-level. In contrastive learning, a key task is to select appropriate samples as the query, positive keys, and negative keys. The selection of query and positive key is relatively straightforward, but the crucial part lies in choosing suitable negative keys so that the final contrastive learning result can be as accurate as possible. Herein, we introduce the concept of entropy to tackle this sampling problem.

Entropy is a metric of the uncertainty or randomness of data or a probability distribution. The entropy of the probability distribution of each pixel can be used as a measure to determine the uncertainty or confusion of the prediction. If a pixel’s Softmax probability entropy value is low, it means that the model is very certain about regarding its prediction that the pixel belongs to a specific category. Conversely, if a pixel’s Softmax probability entropy value is high, it means that the model is uncertain about regarding its class prediction for that pixel. Based on this, we establish an entropy threshold and select pixel samples with an entropy value lower than this threshold as the negative key. In this way, we can ensure to some extent that the selected negative key is less likely to be a misclassified positive key, thus helping to enhance the precision of contrastive learning. The process of contrastive learning is shown in Figure 2.

In this paper, the contrastive learning loss used is as follows:(8)$$L_{con}=-\frac{1}{C\times M}\sum_{c\in C}\sum_{i\in M}log\frac{e^{\left(z_{ci},z_{ci}^{+}/\tau \right)}}{e^{\left(z_{ci},z_{ci}^{+}/\tau \right)}+\sum_{j\in J}e^{\left(z_{ci},z_{cij}^{-}/\tau \right)}}$$

*C* represents the aggregate count of segmentation categories, *M* represents the number of queries and positive keys, and *J* represents the number of negative keys; z=r∘h(x); τ represents the temperature coefficient.

Contrastive learning is performed on the pixel features of the pseudo-label image, and the overall strategy can be summarized as follows: (a) selecting a query; (b) selecting a positive key; (c) selecting a negative key. At the same time, we define the entropy value of pixels.
(9)$$E_{ij}=-\sum_{c\subset C}S_{ij}\left(c\right)logS_{ij}\left(c\right)$$
where $S_{ij}$ represents the softmax probability of the *j*-th pixel in the *i*-th image being of class *c*. To assist in selecting the negative key, we set a threshold α and also define the key-value.

When perturbing the features, the teacher network undergoes weak perturbation, while the student network undergoes strong perturbation. Therefore, we mostly select the features generated by the teacher network as our queries. The definition of query is as follows.
(10)$$Q_{c}^{s}=1\left({\hat{y}}_{ij}=c\right)\left(r\circ \left(h\left(x_{ij}\right)+\delta \right)\right)$$
(11)$$Q_{c}^{t}=1\left({\hat{y}}_{ij}=c\right)\left(r\circ \left(h\left(x_{ij}\right)\right)\right)$$

Subject to $E_{ij}<\alpha$. We generate 80% of all queries from samples in the teacher network, and the remaining 20% are sampled from the student network. For the *i*-th labelled image, $x_{ij}$ is the *j*-th pixel, where ${\hat{y}}_{ij}$ is the predicted label generated by the network. 1() is the indicator function.

The selection rule for the positive key $P_{c}$ is consistent with the query. After selecting the query and positive key, the negative keys $N_{c}$ are randomly sampled from the remaining features, and also need to satisfy the entropy threshold condition, defined as follows:(12)$$N_{c}∼\mathrm{Uniform}\left(z\setminus Q_{c},P_{c}\right)\;\;and\;\;E_{ij}<\alpha$$

After selecting the final key values, the contrastive loss $L_{con}$ can be calculated.

The final loss update of the model in this paper is:(13)$$L=L_{sup}+L_{unsup}+L_{con}$$

### 3.2. Datasets

#### 3.2.1. iSAID

In this paper, we employ the iSAID dataset [49] to evaluate the efficacy and performance of our proposed method for semantic segmentation. The iSAID dataset is a large-scale and challenging benchmark in the field of remote sensing, containing a diverse collection of 15 object categories and a total of 2806 high-resolution aerial images. To facilitate the experimental process, the dataset is divided into a training set with 1411 images and a test set comprising 458 images. During the training phase, we adopt a data augmentation strategy that involves randomly cropping the images to a fixed size of 512×512 pixels.

#### 3.2.2. Potsdam

We also employ the Potsdam dataset [50]. The Potsdam dataset, a benchmark dataset in the realm of remote sensing semantic segmentation, is derived from the Potsdam region in Germany. It is a collaborative product of the International Society for Photogrammetry and Remote Sensing (ISPRS) and the German Society for Photogrammetry, Remote Sensing, and Geoinformation (DGPF). The dataset comprises high-resolution aerial images with a spatial resolution of 5 cm per pixel, captured using an UltraCamXp large-format digital aerial camera. Each image in the dataset has a larger dimension compared to the Vaihingen dataset, covering an area of approximately 1000×1000 meters, providing more detailed ground information. In the Potsdam dataset, there are a total of 24 annotated images used as the training set and 14 test images used as the test set. During the training phase, we randomly cropped the images to a size of 512×512 pixels. Additionally, the image augmentation method used for the Potsdam dataset is the same as that used for the iSAID dataset.

### 3.3. Evaluation Metrics

The assessment measure employed here is the Mean Intersection over Union (mIoU). IoU, a commonly adopted metric for tasks of semantic segmentation, gauges the overlap extent between predicted segments and their respective ground truths. This metric provides a reliable means of assessing the performances of segmentation algorithms, as it takes into account both the false positives and false negatives, thereby offering a comprehensive view of the model’s accuracy. In the context of semantic segmentation, the IoU is often reported as the mean IoU (mIoU), which is the average IoU across all classes present in the dataset.
(14)$$IoU=\frac{TP}{TP+FP+FN}$$
where TP stands for the count of true positives, TN is the number of true negatives, FP symbolizes the quantity of false positives, and FN is the number of false negatives.

The $F_{1}$ score is the harmonic mean of precision and recall, serving as a comprehensive performance evaluation metric. The $F_{1}$ score simultaneously considers the model’s Precision and Recall, being sensitive to both false positives and false negatives. Especially in cases of class imbalance, the $F_{1}$ score can provide a more accurate performance measurement. The calculation formula for the $F_{1}$ score is as follows:(15)$$F_{1}=\frac{2\times Precision\times Recall}{Precision+Recall}$$
(16)$$Precision=\frac{TP}{TP+FP}$$
(17)$$Recall=\frac{TP}{TP+FN}$$

By using both the $F_{1}$ score and IoU, we can simultaneously evaluate the model’s accuracy in classification prediction and spatial localization, providing a more comprehensive evaluation of model performance.

### 3.4. Implementation Detail

The network used in the experiment is Deeplab V3+, with ResNet-101 as the backbone. Images are randomly cropped to a size of 512×512 before being input into the training network, with a batch size of 1 and a training epoch of 200. The learning rate is set to a decayed learning rate, $lr(1-iter/iter_{max}{)}^{0.9}$. The stochastic gradient descent (SGD) optimizer is used, with an initial learning rate set to 0.01 and a learning rate decay of 0.0005. The dataset is divided into 1/2, 1/4, 1/8, and 1/16 labelled images, with the remaining images being used as unlabelled images for training the model. The weight of the auxiliary loss is set to 0.4. When updating the parameter $\theta_{t}$, the EMA smoothing factor γ is set to 0.99.

## 4. Results and Discussion

### 4.1. Comparison Experiments

#### 4.1.1. iSAID

Following the general dataset ratio settings in semi-supervised semantic segmentation, we use ratios of labelled to unlabelled datasets of 1/8, 1/4, and 1/2, as well as conducting experiments using the full dataset. For comparative experiments, all datasets used are kept entirely consistent. To mitigate the impact of randomness, we repeat each experiment under the same settings three times and take the average.

In terms of contrast method selection, we chose the classic Mean Teacher (MT) method for natural scenes, as well as the methods with superior performing methods, RanPaste and GCT. In the field of remote sensing images, we selected recently proposed methods that have shown good performances in semi-supervised semantic segmentation for remote sensing images: ICNet and PICS. The experimental results compared with these methods are shown in Table 1. At the same time, we used bold fonts to represent the best evaluation metrics in this series of comparative experiments. It can be seen that our method achieved the best performance in terms of the mIoU metric on 1/8, 1/4, and 1/2 dataset proportions, with respective results of 42.65%, 45.08%, and 49.11%. Though our metrics on the complete dataset didn’t reach the topmost rank, they are competitively close to the highest-performing PICS-I method—with our method’s mIoU trailing by merely 0.45%. Furthermore, our performance outstrips all of the other methods analyzed in the study.

The iSAID dataset consists of 15 categories, and within these 15 categories, the segmentation difficulty varies. For example, categories like Baseball Diamond and Tennis Court demonstrate better segmentation results, whereas under the same conditions, categories like Helicopter and Bridge prove to be much more difficult to segment. The mIoU for specific categories is shown in Table 2, where the dataset ratio is 1/4. From the results, it can be seen that in the difficult categories, our method has a distinct advantage over other methods, with a clear segmentation advantage in categories like Helicopter and Bridge. These results demonstrate the effectiveness of our improvements in feature space. Specific analyses will be elaborated in conjunction with the upcoming ablation studies.

The visualization of the segmentation results on the iSAID dataset is shown in Figure 3. What we’ve presented here are the results of the segmentation, performed on a quarter of the dataset. To better demonstrate the advantages of our segmentation, we have chosen to present segments that are challenging to separate, and we contrast these with the results from the PICS method. We chose the PICS method specifically because it exhibits the best performance in experiments aside from our own. As is evident, our method is the only one that can identify helicopters. Moreover, when considering other objects, our method visibly provides superior segmentation results.

#### 4.1.2. Potsdam

Remote sensing scenes are diverse and complex. The categories included in the iSAID dataset cannot fully represent remote sensing images. Therefore, we need additional datasets to demonstrate the universality of our method on remote sensing images. Here, we choose the Potsdam dataset, which contains a large number of buildings and streets. The experimental setup is the same as the iSAID dataset. Table 3 shows the Potsdam segmentation performance of different methods under different dataset proportions.

From the results, it can be seen that our method also achieved the best overall performance on the Potsdam data. All comparison methods on the Potsdam dataset have superior performance. Although our method achieved the best performance in mIoU, with results of 79.33%, 85.01%, and 85.93% under the dataset proportions of 1/8, 1/4, and 1/2, there is no noticeable gap. This is because the targets in the Potsdam dataset are quite distinct and easy to classify. The IoU data of specific categories under the dataset proportion of 1/4 is shown in Table 4, where our method demonstrated the best performance in the Buildings, Trees, and Cars categories.

The visualization of the segmentation on the Potsdam dataset is shown in Figure 4.

### 4.2. Ablation Study

In this section, to delve deeper into the impact of each module on the overall method, and to conduct accurate quantitative assessments, we decided to perform a detailed ablation study on each module. In this series of experiments, we chose to use the iSAID dataset. The reason for this choice is that, compared to the Potsdam dataset, the images in the iSAID dataset are more challenging to accurately segment, and the differences in segmentation metrics can more effectively reflect the practical utility of each module. To ensure the fairness and consistency of the experiments, we uniformly used a 1/4 dataset ratio in all ablation studies.

We will begin by conducting a detailed ablation study on the overall modules. In our approach, there are several key modules that can be ablated, their functions and roles are as follows:

Feature Disturbed Module (FDM): The primary function of this module is to enhance the disturbance of features encoded by the student and teacher networks. Through this method, we can improve the robustness and the generalization ability of the model while ensuring network performance.

Contrastive Learning Module ($L_{c}$): The main task of this module is to deeply optimize the feature space using a contrastive learning strategy. In this module, we have set an entropy threshold to filter negative keys, aiming to enhance the effectiveness and precision of contrastive learning. To verify the effectiveness of this strategy, we also need to conduct a detailed ablation study on this module.

Through such ablation studies, we can gain a deeper understanding and evaluation of the specific contributions and impacts of each module to the overall method, thereby better optimizing our model and approach. The results of the ablation study are shown in Table 5. The overall results of the ablation experiments confirm that each module positively contributes to the performance of the model. We ablated all modules, resulting in a basic model that served as our baseline for comparison. During the ablation of the contrastive learning module, we used the basic cross-entropy loss as the loss function.

On this basic model, we incrementally added different modules, starting with the FDM module. The results showed that just by adding this one module, the model’s mIoU increased by 2.32%. This result validates the effectiveness of our strategy of transferring consistency regularization criteria to feature disturbance. After separately adding the contrastive learning module, the mIoU increased by 1.11%. On this basis, we added an entropy threshold for auxiliary key-value screening, which further improved the mIoU by 0.69%. Although we added the contrastive learning module alone without introducing any feature-level disturbance, it is undeniable that contrastive learning optimizes the feature space, especially in the case of remote sensing images. Finally, we combined all of the modules to form our final model. This model achieved a mIoU increase of 3.85%. This result fully demonstrates the effectiveness of our modular design and the significant contribution of each module to the performance of the model.

To more clearly demonstrate the effects of our ablation study, we have additionally incorporated t-SNE visualizations using the Potsdam dataset, as illustrated in Figure 5. From these visualizations, it is evident that the feature disturbed module contributes to optimizing the feature space. Furthermore, employing contrastive learning on top of FDM significantly enhances the optimization of the feature space. This combination of techniques clearly delineates the improvements in feature representation and model performance.

Within each module, we have made meticulous configurations. For example, in the feature disturbed module, we opted for a combination of strong and weak disturbances, and determined how to set the entropy threshold to achieve the best result. To demonstrate and to validate the rationality of these settings, we conducted a series of thorough and detailed ablation experiments.

#### 4.2.1. Ablation Study of FDM

The proposal of FDM was inspired by the optimization assistance of strong and weak enhancements at the image level. In its construction, we used a strong enhancement: VAT, and a weak enhancement: feature jittering. We set up the ablation experiment for FDM as follows: the teacher network and the student network each generate VAT for feature perturbation; the teacher network uses the feature jittering method, and the student network uses VAT, as in our original experimental setup; both the teacher network and the student network use feature jittering for feature perturbation. The experimental results are shown in Table 6.

The results from the Table 6 indicate that the simultaneous use of VAT and feature jittering achieved the best performance. However, when using VAT or feature jittering alone, the mIoU decreased by 2.82% and 1.63%, respectively. When using VAT alone, the features of both networks underwent significant perturbations. Although contrastive learning was used to match the feature space, the excessive randomness led to a more chaotic and disordered feature space, increasing the learning difficulty and degrading the performance. In contrast, when using feature jittering alone, the final mIoU score was quite close to the result without feature perturbations, indicating a certain degree of performance improvement. This is because when we applied feature jittering, the noise we set caused only slight perturbations to the features. Therefore, the difference between the perturbed feature maps and the original features was not significant. At the same time, it added a certain degree of randomness, which helped to maintain feature stability after network training, thereby improving model performance to some extent.

Our original experimental setup, namely the use of both VAT and feature jittering, showed markedly better performance than the above two scenarios. This validates the reasonableness of our method setup and also proves that using the results generated by weak augmentation to assist the results generated by strong augmentation in migrating to the feature space is equally applicable.

#### 4.2.2. Ablation Study of the Entropy Threshold

During the process of feature key-value selection, we employed an entropy threshold as an auxiliary tool. The setting of this entropy threshold requires careful selection, for which we conducted a series of exhaustive ablation experiments. Our method involves calculating the entropy of all pixel points and then selecting a certain percentile as the entropy threshold. The advantage of this approach is that the threshold can adjust according to the overall change in entropy, rather than solely relying on a fixed threshold. This method ensures that our feature key-value selection is more flexible and adaptable, and better equipped to handle different data distributions and complexities. We experimented with multiple percentiles for the entropy threshold, including 0.4, 0.6, 0.7, 0.8, and 0.99. The final mIoU results are displayed in Table 7.

The experimental results show that selecting the 0.7 percentile as the entropy threshold yields the best results. When we filter key-values, we choose those that are below a certain entropy threshold, resulting in more accurate key-values. If the threshold is set too high, such as at the 0.8 and 0.99 percentile entropy values, many key-values that the model is uncertain about may still be selected, leading to classification errors, with their mIoUs being 0.14% and 0.49% lower than the optimal setting, respectively. This is because an excessively high threshold might affect the selection of positive keys, potentially misclassifying latent postive keys as negative keys, thus leading to a decrease in results. Conversely, if the key-value selection is too low, such as 0.4 and 0.6, it does not achieve better results than 0.7, with the mIoUs being 0.37% and 0.06% lower than the optimal setting, respectively. This is because, although some pixel points have high degrees of uncertainty, they may contain valuable information such as the boundary information of the positive category, which has high predictive uncertainty but is important for model training. Therefore, blindly lowering the threshold does not improve performance. Based on this, we chose a threshold percentile of 0.7, achieving the best results in model training and prediction.

## 5. Conclusions

In this paper, we have delved into the unique characteristics of remote sensing images and have introduced an innovative semi-supervised semantic segmentation framework. This framework synergistically integrates consistency regularization methods and contrastive learning techniques, significantly advancing the field of semi-supervised semantic segmentation. Initially, we establish a unique learning paradigm for consistency regularization, introducing perturbations at the feature level, and enhancing both strong and weak perturbations to maintain the stability of features after training. Secondly, for the feature space of the images, we introduce contrastive learning to achieve better partitioning and classification. By leveraging the entropy threshold to assist in feature selection, we are able to select more accurate positive keys and negative keys, making the results of contrastive learning more precise. Our method surpasses the state-of-the-art techniques on the Potsdam and iSAID datasets. Concerning the entropy threshold setting, our current model utilizes a static value. However, we recognize the potential for performance optimization through a dynamically adapting entropy threshold that evolves in tandem with model training. This insight forms the basis for our future research direction, where we aim to develop and validate an effective dynamic entropy threshold model, further pushing the boundaries of semi-supervised semantic segmentation in remote sensing imagery.

## Figures and Tables

**Figure 1 sensors-24-00730-f001:**
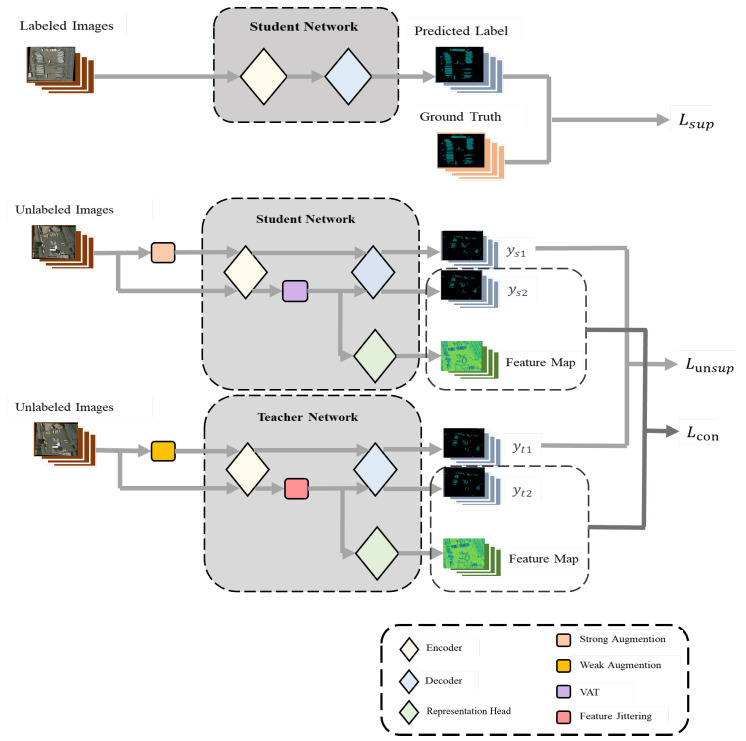
Overview of the proposed method framework. This framework includes a student network and a teacher network. Both networks share the same architecture. The optimization of the framework is based on three loss functions. $L_{sup}$ is generated from the predicted labels of the annotated images and the ground truth; $L_{unsup}$ is calculated from the predicted labels of two unannotated images that have undergone image enhancement; $L_{con}$ is generated from the predicted labels of two features-perturbed images calculated through contrastive learning.

**Figure 2 sensors-24-00730-f002:**
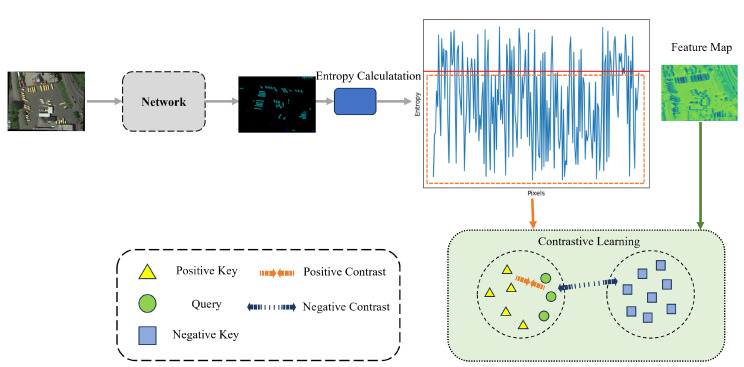
Entropy-assisted contrastive learning. The softmax probabilities of the labels predicted by the network pass through an entropy calculation module, obtaining the entropy value for each pixel. An entropy threshold is set, as shown by the red line in the figure. We select the pixels and their features below the entropy threshold as the key values for contrastive learning.

**Figure 3 sensors-24-00730-f003:**
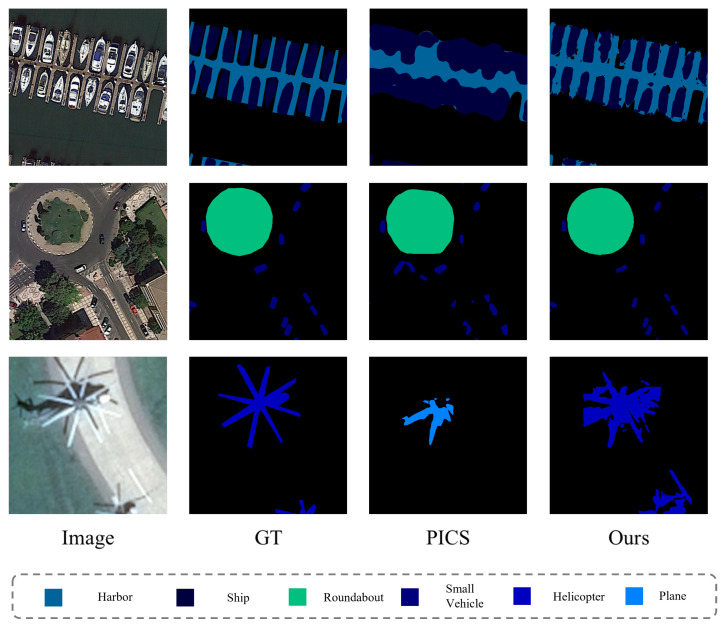
iSAID dataset segmentation result visualization image.

**Figure 4 sensors-24-00730-f004:**
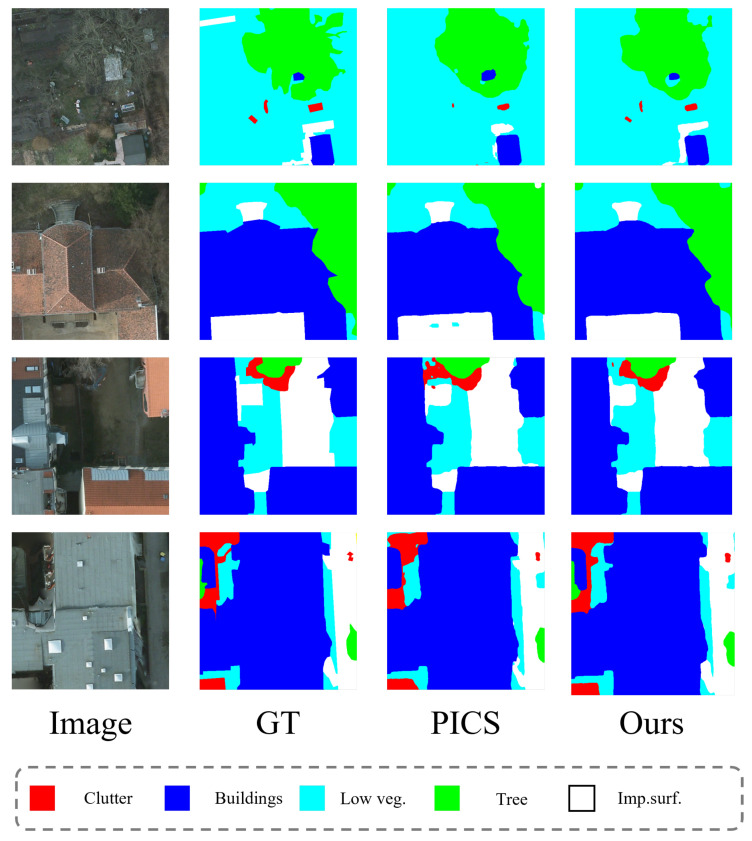
Potsdam dataset segmentation result visualization image.

**Figure 5 sensors-24-00730-f005:**
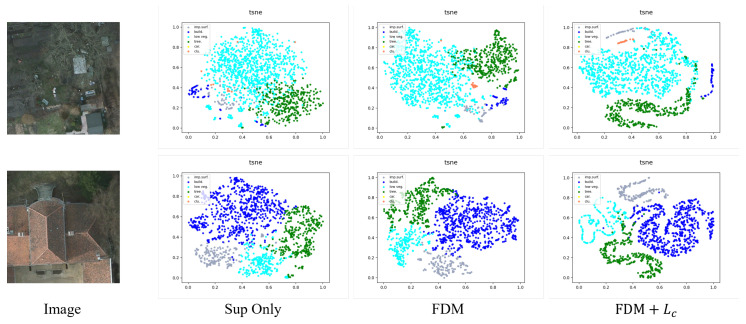
t-SNE visualization of feature spaces on test images of Potsdam dataset.

**Table 1 sensors-24-00730-t001:** Comparisons with the SOTA methods evaluated on the iSAID dataset.

Method	1/8			1/4			1/2			Full	
	mIoU(%)	m$F_{1}$(%)		mIoU(%)	m$F_{1}$(%)		mIoU(%)	m$F_{1}$(%)		mIoU	m$F_{1}$(%)
MT [8]	39.76	56.90		41.91	59.07		45.33	62.38		49.97	66.64
RanPaste [51]	41.11	58.27		42.38	59.53		47.06	64.00		50.29	66.92
ICNet [17]	42.14	59.29		42.67	59.82		46.80	63.76		50.65	67.24
GCT [29]	40.09	57.23		41.03	58.19		46.91	63.86		50.74	67.32
PCIS [5]	42.63	59.78		44.28	61.38		48.91	65.69		**53.90**	**70.05**
(ours )	**42.65**	**59.80**		**45.08**	**62.15**		**49.11**	**65.87**		53.45	69.66

**Table 2 sensors-24-00730-t002:** Comparisons with the SOTA methods evaluated on the iSAID dataset.

Method	SH	RA	BD	TC	BC	GTF	BR	LV
	SV	HC	SP	ST	SBF	PL	HA	mIoU(1/4)
MT [8]	47.53	**56.29**	63.27	64.79	27.84	30.32	9.03	62.49
	33.68	4.43	69.40	**31.17**	40.57	39.84	47.99	41.91
RanPaste [51]	46.63	50.58	54.98	69.35	27.99	29.39	9.36	**68.12**
	30.21	9.14	66.44	26.68	52.15	46.20	48.44	42.38
ICNet [17]	51.49	47.56	**66.43**	65.76	24.75	28.96	9.04	65.28
	35.72	8.89	65.97	21.40	48.98	**49.07**	50.72	42.67
GCT [29]	49.14	49.11	44.94	67.88	24.61	25.60	11.63	58.47
	35.07	10.77	56.48	25.91	**65.00**	42.02	48.81	41.03
PCIS [5]	50.20	49.32	55.76	70.42	29.75	28.16	15.13	65.46
	34.17	13.65	68.60	26.65	53.76	47.47	**55.76**	44.28
(ours)	**51.12**	49.45	55.55	**71.48**	**30.39**	**29.57**	**19.23**	65.81
	34.36	**18.83**	**68.54**	27.01	54.64	47.86	52.43	**45.08**

**Table 3 sensors-24-00730-t003:** Comparisons with the SOTA methods evaluated on the Potsdam dataset.

Method	1/8			1/4			1/2	
	mIoU(%)	m$F_{1}$(%)		mIoU(%)	m$F_{1}$(%)		mIoU(%)	m$F_{1}$(%)
MT [8]	78.94	88.23		84.52	91.61		85.10	91.95
Ranpaste [51]	77.95	87.61		84.01	91.31		85.23	92.03
ICNet [17]	78.61	88.02		83.59	91.06		85.07	91.93
GCT [29]	78.80	88.14		84.17	91.40		85.22	92.02
PCIS [5]	78.95	88.24		84.66	91.69		85.36	92.10
(ours)	**79.33**	**88.47**		**85.01**	**91.90**		**85.93**	**92.43**

**Table 4 sensors-24-00730-t004:** Comparisons with the SOTA methods evaluated on the Potsdam dataset.

Method	Imp.surf.	Buildings	Low veg.	Tree	Car	mIoU (1/4)
MT [8]	90.56	82.36	79.95	80.57	89.16	84.52
RanPaste [51]	91.31	82.39	78.85	79.27	87.72	84.01
ICNet [17]	90.94	82.41	80.60	80.52	87.47	84.39
GCT [29]	**91.42**	**80.94**	78.61	80.76	88.25	84.17
PCIS [5]	91.27	81.82	**80.68**	80.58	88.95	84.66
(ours)	91.22	**82.84**	80.45	**81.33**	**89.21**	**85.01**

**Table 5 sensors-24-00730-t005:** Ablation study on the effectiveness of components in our method.

FDM	$L_{c}$	Entropy Threshold	mIoU(%)	m$F_{1}$(%)
			41.23	58.39
✓			43.55	60.68
	✓		42.34	59.49
	✓	✓	43.03	60.17
✓	✓		44.18	61.28
✓	✓	✓	**45.08**	**62.15**

**Table 6 sensors-24-00730-t006:** Ablation study on the effectiveness of FDM.

Feature Perturbation	VAT & VAT	FJ & VAT	FJ & FJ
mIoU (%)	42.26	45.08	43.45
m$F_{1}$ (%)	59.41	62.15	60.58

**Table 7 sensors-24-00730-t007:** Ablation study on the effectiveness of entropy threshold.

Entropy Threshold	0.4	0.6	0.7	0.8	0.99
mIoU (%)	44.71	45.02	**45.08**	44.94	44.59
m$F_{1}$ (%)	61.79	62.09	62.15	62.01	61.68

## Data Availability

Data are available in a publicly accessible repository that does not issue DOIs and are provided by third parties. The datasets that we use are available at https://captain-whu.github.io/iSAID/index.html (accessed on 1 June 2023) and https://seafile.projekt.uni-hannover.de/f/429be50cc79d423ab6c4/ (accessed on 1 June 2023).

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
