# Peer review of "Enhancing Semi-Supervised Semantic Segmentation of Remote Sensing Images via Feature Perturbation-Based Consistency Regularization Methods"

_sensors, 2024, doi:10.3390/s24030730_

Round 1

Reviewer 1 Report

Comments and Suggestions for Authors

The manuscript proposes a semi-supervised semantic segmentation method with consistency regularization and contrastive learning. By enhancing perturbations at the feature level and  utilizing contrastive learning at the feature level, the proposed method works better than baselines. Here are some detailed comments:

1.     In the section 3.3, there are only one evaluation metrics. To enhance persuasiveness, the evaluation metrics should be more in the table1 and table 3 at least. The meaning of TP, FP and FN is not explained.

2.     The teacher model weight is updated by EMA, therefore the formula for EMA should be written in section 3 and the parameter values in the EMA formula should be explained in section 3.4

3.     The position of the figures in the paper should be adjusted, as much as possible near the position of the cited charts. The figure should indicate the class of different colors.

4.     For the ablation experiment section, if convenient, it is recommended to add a t-sne image for visual display of the results

Comments on the Quality of English Language

 In the first sentence of the second paragraph of section 1, the expression of this sentence is not accurate enough, “of images” should be changed to “of labeled images”.

Reviewer 2 Report

Comments and Suggestions for Authors

The authors propose a semi-supervised semantic segmentation framework that combines consistency regularization and contrastive learning. For consistency regularization, based on MT, the proposed method introduces feature-level disturbances, allowing the model to pay attention to the feature level of the image during the learning process. Simultaneously, to maintain consistency after feature perturbation, authors employ contrastive learning for feature-level learning and utilize entropy threshold to assist contrastive learning.

I recommend publishing this work, but the authors need to address the following issues:

1 There are some errors of logic and expression in the article. Please revise the following sentences:

a)        “According to the principle of semi-supervised learning, labeled images represent only a small fraction of all training images, |Dl | |Du|”. |Dl | >> |Du| VS Dl | << |Du|.

b)        “Remote sensing images have high resolution, with the number of pixels in a single image far exceeding that in other fields. To address this issue, in recent years, some researchers have started to conduct research on semi-supervised semantic segmentation.” What does “this issue” refer to?

2 In the Introduction section, please use more concise statements to summarize the challenges of semantic segmentation of remote sensing images and disadvantages of existing methods. Specifically, the paragraphs 2-6 of the Introduction section can be concluded as one or two paragraphs with coherent sentences.

3 The novelty of the FDMT module is not clearly explained. Specifically, please provide a detailed explanation of the distinctions between the FDMT module and the traditional Mean Teacher model.

4 Please demonstrate the rationality of following settings: “In the student network part, unlabeled image follows two different processing streams: one with strong data augmentation, and one without data augmentation.” “As for the teacher network, unlabeled image is also divided into two processing steps: weak data augmentation and no augmentation.”  What are the reasons for these setups? Is this method being proposed for the first time? If not, please give the corresponding reference.

5 The proposed method utilizes a constant threshold in the part of Entropy-assisted contrastive learning, but how to choose the best value as the entropy threshold? Would it be better to design an adaptive threshold for each pixel (or a dynamic threshold)?

6 More recent works should be cited and discussed. Please read the following related works and discussed them in your work. EA-TGRS PICS_ Paradigms Integration and Contrastive Selection for Semi-Supervised Remote Sensing Images Semantic Segmentation 2022-GRSL Semi-Supervised Semantic Segmentation of Remote Sensing Images With Iterative Contrastive Network 2023-PR Learning Discriminative Feature Representation with Pixel-level Supervision for Forest Smoke Recognition.

7 Please provide details on experimental setup for the Potsdam dataset. Specifically, how many images are divided into training set and test set? During the training phaseis there a data enhancement strategy adopted like iSAID dataset?

8 Please give explicit citations in the body of the article for the 45-78 references.

Comments on the Quality of English Language

There are some errors of logic and expression in the article. More details are provided in the Reviewer Comments.

Round 2

Reviewer 2 Report

Comments and Suggestions for Authors

The revised manuscript has effectively responded to the raised comments. Based on this comprehensive revision, I recommend publishing this paper.